# FluenceFormer: Transformer-Driven Multi-Beam Fluence Map Regression for Radiotherapy Planning

**Ujunwa Mgboh**[1]        UJUNWA.MGBOH@WAYNE.EDU

**Rafi Ibn Sultan**[1]        RAFIS@WAYNE.EDU

**Joshua Kim**[2]        JKIM8@HFHS.ORG

**Kundan Thind**[2]        KTHIND1@HFHS.ORG

**Dongxiao Zhu**[1]        DZHU@WAYNE.EDU

[1] *Wayne State University, Detroit, MI*

[2] *Henry Ford Health, Detroit, MI*

**Editors:** Accepted for publication at MIDL 2026

## Abstract

Fluence map prediction is central to automated radiotherapy planning but remains an ill-posed inverse problem due to the complex relationship between volumetric anatomy and beam-intensity modulation. Convolutional methods in prior work often struggle to capture long-range dependencies, which can lead to structurally inconsistent or physically unrealizable plans. We introduce **FluenceFormer**, a backbone-agnostic transformer framework for direct, geometry-aware fluence regression. The model uses a unified two-stage design: Stage 1 predicts a global dose prior from anatomical inputs, and Stage 2 conditions this prior on explicit beam geometry to regress physically calibrated fluence maps. Central to the approach is the **Fluence-Aware Regression (FAR)** loss, a physics-informed objective that integrates voxel-level fidelity, gradient smoothness, structural consistency, and beam-wise energy conservation. We evaluate the generality of the framework across multiple transformer backbones, including Swin UNETR, UNETR, nnFormer, and MedFormer, using a prostate IMRT dataset. FluenceFormer with Swin UNETR achieves the strongest performance among the evaluated models and improves over existing benchmark CNN and single-stage methods, reducing Energy Error to **4.5**% and yielding statistically significant gains in structural fidelity ($p < 0.05$). Code available at: https://github.com/UJUNWAMGBOH/FluenceFormer

**Keywords:** Transformer-based models, Fluence map prediction, Cancer diagnosis, Cancer treatment, Swin UNETR, Medical imaging, Radiation therapy.

## 1. Introduction

Deep learning has revolutionized medical image analysis, driving major advances in segmentation (Hatamizadeh et al., 2022, 2021; Li et al., 2025b,a, 2023), classification (Chowdary and Yin, 2024; Huang et al., 2024; Yue and Li, 2024; Li et al., 2019), and automated treatment planning (Heilemann et al., 2025; Bakx et al., 2023; Nemoto et al., 2025). In Intensity-Modulated Radiation Therapy (IMRT), optimal dose delivery requires balancing tumor coverage with organ-at-risk (OAR) protection (Bortfeld, 2006; Saw et al., 2001). This balance is physically realized through multi-beam fluence maps, spatial intensity patterns that modulate the linear accelerator's multileaf collimator (MLC). Automating fluence prediction offers the potential to drastically reduce planning time, improve reproducibility,

and standardize clinical quality (Wang et al., 2021; Yuan et al., 2022; Wang et al., 2020; Vandewinckele et al., 2022; Li et al., 2020; Ma et al., 2020).

Most learning-based radiotherapy approaches focus on segmentation (Hatamizadeh et al., 2022, 2021) or voxel-wise dose prediction (Gheshlaghi et al., 2024; Hu et al., 2023; Wang et al., 2024; Wen et al., 2023), relying on subsequent inverse planning optimization to generate deliverable parameters. Direct learning of fluence maps provides a streamlined alternative by mapping anatomy directly to machine parameters (Arberet et al., 2025; Cai et al., 2024). However, this mapping is inherently inadequate: a specific internal dose distribution can be achieved by infinite combinations of beam angles and intensities. Convolutional Neural Networks (CNNs), which dominate current research (Wang et al., 2021; Yuan et al., 2022; Wang et al., 2020; Vandewinckele et al., 2022; Li et al., 2020; Ma et al., 2021, 2020), struggle to resolve this ambiguity. Their limited receptive fields restrict the modeling of long-range anatomical dependencies and the global cross-beam correlations necessary for physically realizable modulation.

Vision Transformers (ViTs) present a natural solution, as self-attention mechanisms enable global context modeling and anisotropic feature reasoning (Hatamizadeh et al., 2022, 2021; Zhou et al., 2021; Chowdary and Yin, 2024). These capabilities align well with fluence estimation, where beam modulation is governed by volumetric tissue geometry and angular interactions. Despite this, transformer-based fluence prediction remains underexplored. Our recent preliminary study (Mgboh et al., 2025) demonstrated early promise but had key limitations: it relied on a single backbone architecture, leaving generalization across transformer families untested, and employed a simple pixel-wise loss without physics-informed constraints for spatial or energetic consistency.

We introduce **FluenceFormer**, a backbone-agnostic transformer framework for direct beam-specific fluence prediction. The model adopts a novel two-stage architecture: *Stage 1* predicts a global dose distribution from anatomical inputs, serving as a structural prior; *Stage 2* conditions this dose estimate on explicit beam geometry to regress physically calibrated fluence maps. This hierarchical design mirrors the clinical workflow, where prescribed doses guide beam optimization, and effectively resolves the geometric ambiguity of direct prediction. To ensure clinical deliverability, we propose the **Fluence-Aware Regression (FAR)** loss, a physics-informed objective that enforces voxel-level fidelity, gradient smoothness, and strict energy conservation (i.e., Monitor Unit consistency). Finally, we provide a comprehensive evaluation against strong CNN and single-stage baselines, as well as segmentation-style baselines across four transformer backbones (UNETR (Hatamizadeh et al., 2022), Swin UNETR (Hatamizadeh et al., 2021), nnFormer (Zhou et al., 2021), and MedFormer (Gao et al., 2022)), and present a complete analysis of FluenceFormer with the proposed FAR loss across multiple backbones, demonstrating its architecture-agnostic capabilities.

## 2. Methodology

### 2.1. Problem Definition

**Data and notation.** We consider per-patient datasets consisting of a CT volume and corresponding anatomical contours, defined on a voxel grid with depth $D$ and in-plane resolution $H \times W$. Each case is associated with a ground-truth dose distribution and $B$

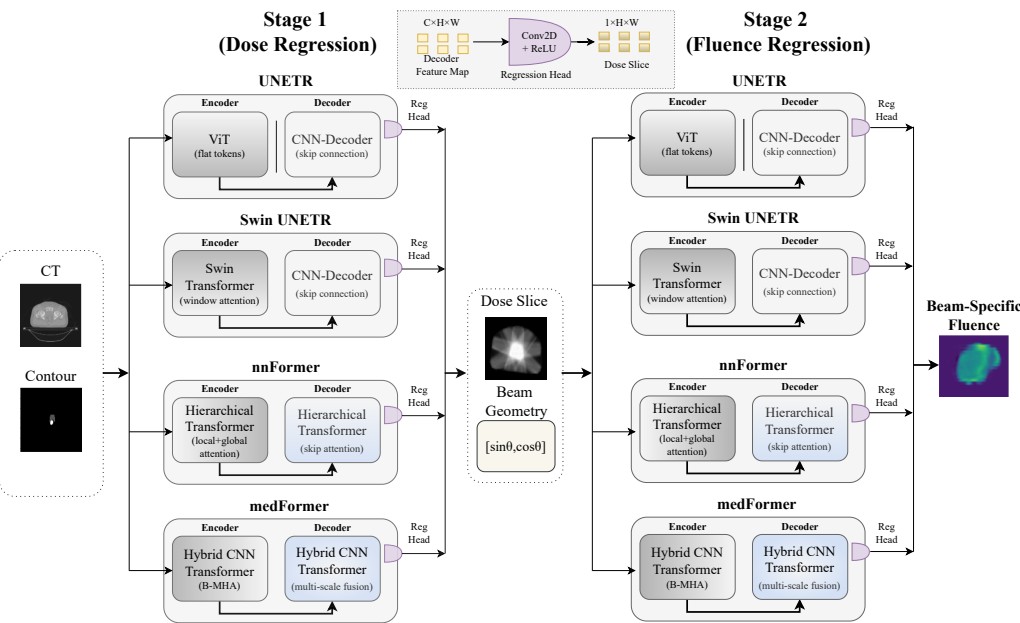

Figure 1: Overview of the **FluenceFormer** framework. Stage 1 performs slice-wise dose regression from anatomical inputs. Stage 2 conditions the predicted dose on explicit geometric encoding (sin/cos) to resolve beam directionality and regress beam-specific fluence maps. ReLU-based regression heads replace segmentation classifiers to enable the prediction of unbounded physical intensities (Monitor Units).

beam-specific fluence maps. Formally, we denote:

$$C \in \mathbb{R}^{D \times H \times W}, \quad R \in \mathbb{R}^{D \times H \times W}, \quad Y_{\text{dose}} \in \mathbb{R}^{D \times H \times W}, \quad F \in \mathbb{R}^{B \times H \times W},$$

where $C$ represents the CT volume, $R$ the anatomical contour masks, $Y_{\text{dose}}$ the dose distribution, and $F$ the beam-specific fluence maps.

**Slice-level formulation.** Let $z \in \{1, \ldots, D\}$ index the axial slices. For Stage 1, the model receives a two-channel slice input comprising the anatomy and contours:

$$x_z = [\, C_z, \, R_z \,] \in \mathbb{R}^{2 \times H \times W}, \tag{1}$$

and predicts the corresponding dose slice $y_z = Y_{\text{dose},z}$.

## 2.2. FluenceFormer: The Framework

FluenceFormer (illustrated in Figure 1) follows a two-stage formulation that mirrors the clinical workflow in radiation therapy, where dose objectives are defined before beam fluence optimization.

### 2.2.1. STAGE 1: DOSE REGRESSION

**Backbone.** Stage 1 adapts transformer-based segmentation architectures for continuous radiotherapy regression. We adopt four representative backbones under one single framework: UNETR, Swin UNETR, nnFormer, and MedFormer, utilizing their implementations

as flexible feature extractors. Stage 1 predicts the dose distribution slice-by-slice, aggregating local anatomical context to model how patient geometry modulates radiation transport. This intermediate dose map encodes spatial context and treatment intent, serving as a structural prior for Stage 2. Each axial slice $x_z$ is processed to produce a decoder feature map $X_z^{\mathrm{dec}}$.

**Regression head.** To convert these backbones into dose regressors, we employ a pixel-wise regression head as illustrated in Figure 1. A single $1 \times 1$ convolution followed by a Rectified Linear Unit (ReLU) activation generates the dose slice:

$$\hat{y}_z = \mathrm{ReLU}\Big(\mathrm{Conv}_{1\times1}(X_z^{\mathrm{dec}})\Big), \quad \hat{y}_z \in \mathbb{R}_{\geq 0}^{1 \times H \times W}. \tag{2}$$

This design allows the model to predict physical dose values directly without the dynamic range saturation issues associated with sigmoid normalization, while providing a lightweight and stable interface for adapting segmentation-based transformers to continuous regression and maintaining architectural consistency across backbones.

**Training objective.** Stage 1 is optimized using a mean-squared error (MSE) loss for voxel-wise supervision:

$$\mathcal{L}_{\mathrm{S1}} = \frac{1}{|\Omega|} \sum_{x \in \Omega} \big(\hat{y}_z(x) - y_z(x)\big)^2. \tag{3}$$

### 2.2.2. Stage 2: Geometry-Conditioned Fluence Regression

**Input and Conditioning.** As can be seen from Figure 1, stage 2 maps the predicted dose to beam-specific fluence maps. Crucially, unlike standard decoding which assumes a fixed output channel ordering, we explicitly condition the network on beam geometry to resolve the inperfect relationship between scalar dose and directional fluence.

The input $x_z^{(2)}$ is centered on the Stage 1 dose prediction $\hat{y}_z$, supplemented by geometric encoding. For a target beam $b$ with gantry angle $\theta_b$, we generate two spatial maps, $M_{\mathrm{sin}}$ and $M_{\mathrm{cos}}$, filled with values $\sin(\theta_b)$ and $\cos(\theta_b)$ respectively. These are concatenated with the dose to form a 3-channel input:

$$x_z^{(2)} = [\,\hat{y}_z,\, M_{\mathrm{sin}},\, M_{\mathrm{cos}}\,] \in \mathbb{R}^{3 \times H \times W}. \tag{4}$$

This formulation ensures the network relies on the dose distribution to drive intensity modulation, while the geometric maps explicitly disambiguate the directionality of the beam without requiring re-processing of the anatomy.

**Prediction.** The input is processed by the transformer backbone to yield the specific fluence map for beam $b$. To enforce physical non-negativity while preserving the full dynamic range of beam intensities (Monitor Units), we employ a ReLU activation:

$$\hat{F}_z^b = \mathrm{ReLU}\Big(\mathrm{Conv}_{1\times1}(X_z^{\mathrm{dec2}})\Big), \quad \hat{F}_z^b \in \mathbb{R}_{\geq 0}^{1 \times H \times W}. \tag{5}$$

During inference, this process is iterated for all desired beam angles $\{1, \ldots, B\}$ to construct the full multi-beam plan.

**Baseline objective.** Stage 2 is trained using an MSE loss averaged over all sampled beams:

$$\mathcal{L}_{\text{baseline}} = \frac{1}{B|\Omega|} \sum_{b=1}^{B} \sum_{x \in \Omega} \big(\hat{F}_z^b(x) - F_z^b(x)\big)^2. \tag{6}$$

### 2.3. Proposed Loss Function: Fluence-Aware Regression (FAR)

#### 2.3.1. FORMULATION

We define the **Fluence-Aware Regression (FAR)** objective to supervise the prediction of beam-wise fluence maps. The loss comprises four complementary components that jointly enforce pixel-level accuracy, spatial smoothness, structural consistency, and global energy agreement:

$$\mathcal{L}_{\text{FAR}} = \alpha\,\mathcal{L}_{\text{MSE}} + \beta\,\mathcal{L}_{\text{Grad}} + \gamma\,\mathcal{L}_{\text{Corr}} + \delta\,\mathcal{L}_{\text{Energy}}, \quad \alpha, \beta, \gamma, \delta \geq 0. \tag{7}$$

Let $\Omega$ denote the 2D beam's-eye-view domain and $B$ the total number of treatment beams. For each beam $b \in 1, \ldots, B$, the model predicts a fluence map $\hat{F}^{,b}$ that is compared against its ground-truth counterpart $F^{,b}$. The following loss components operate over these beam-wise fluence fields.

**Pixel Fidelity.** We employ MSE to enforce strict intensity agreement at the pixel level. This term provides the foundational supervision for regressing absolute fluence values:

$$\mathcal{L}_{\text{MSE}} = \frac{1}{B|\Omega|} \sum_{b=1}^{B} \sum_{x \in \Omega} \big(\hat{F}^b(x) - F^b(x)\big)^2. \tag{8}$$

**Gradient Smoothness.** Fluence modulation is physically constrained by the mechanical movement of the Multileaf Collimator (MLC), which precludes sharp, high-frequency intensity spikes. We penalize local derivative mismatches to ensure spatially smooth, deliverable transitions, inspired by gradient consistency losses in image super-resolution (Ledig et al., 2017; Zhang et al., 2018):

$$\mathcal{L}_{\text{Grad}} = \frac{1}{B|\Omega|} \sum_{b=1}^{B} \sum_{x \in \Omega} \Big( \|\nabla_x \hat{F}^b(x) - \nabla_x F^b(x)\|_1 + \|\nabla_y \hat{F}^b(x) - \nabla_y F^b(x)\|_1 \Big). \tag{9}$$

**Correlation Consistency.** While MSE minimizes magnitude errors, it does not explicitly enforce shape similarity. This term maximizes the Pearson correlation coefficient $\rho$ to ensure the predicted modulation pattern structurally aligns with the ground truth, independent of absolute scaling (Ferdousi et al., 2021; Wang et al., 2004):

$$\mathcal{L}_{\text{Corr}} = \frac{1}{B} \sum_{b=1}^{B} \big(1 - \rho(\hat{F}^b, F^b)\big). \tag{10}$$

**Energy Conservation.** To guarantee dosimetric validity, the total predicted photon flux must match the prescription. We model the total beam energy as the discrete integral of fluence intensity scaled by the physical pixel area $\Delta A$ (in mm$^2$). The loss penalizes

deviations in total Monitor Units (MUs), rooting the prediction in IMRT physics (Bortfeld, 2006; Zhang and Merritt, 2006; Saw et al., 2001):

$$\mathcal{L}_{\text{Energy}} = \frac{1}{B} \sum_{b=1}^{B} \left| \sum_{x \in \Omega} \hat{F}^b(x) \Delta A - \sum_{x \in \Omega} F^b(x) \Delta A \right|. \tag{11}$$

By incorporating $\Delta A$, this term anchors the regression to the physical output constraints of the treatment machine rather than arbitrary image statistics.

### 2.4. Datasets and Preprocessing

**Datasets.** This study utilized anonymized data from ninety-nine ($n = 99$) prostate cancer patients who underwent intensity-modulated radiation therapy (IMRT) at our institution.

Each case included a planning CT scan, anatomical contours, clinical dose distribution, and nine beam-specific fluence maps. CT images were acquired on a Brilliance Big Bore scanner (Philips Healthcare, Cleveland, OH) at 140 kVp and 500 mAs with $512 \times 512$ in-plane dimensions, corresponding to a spatial resolution of $1.28 \times 1.28$ mm$^2$ and 3 mm slice thickness. All treatment plans were generated in the Eclipse Treatment Planning System (Varian Medical Systems, Palo Alto, CA) using nine equally spaced IMRT fields delivering 70.2–79.2 Gy in 1.8–2.0 Gy fractions. CT, contour, and dose data were exported from Eclipse as DICOM files, while beam fluence maps, originally stored as text-based `.optimal fluence` files, were converted into `NumPy` arrays via custom scripts to ensure compatibility with the learning pipeline.

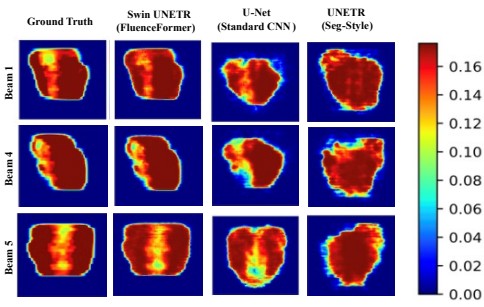

Figure 2: **Qualitative comparison of fluence predictions.** Methods correspond to Table 1.

**Preprocessing.** All volumetric data were resampled to a uniform in-plane grid of $128 \times 128$ pixels using bilinear interpolation. Fluence maps with differing original resolutions were upsampled to the same grid. CT intensities were clipped to a soft-tissue window and normalized to $[0, 1]$. To facilitate physical value regression compatible with the proposed ReLU activation and Energy loss, ground truth dose and fluence maps were not min-max normalized per sample. Instead, they were scaled by a fixed global constant to preserve relative physical intensities (Monitor Units) across the patient population. For Stage 1 dose regression, each CT slice was concatenated with its corresponding multi-channel anatomical masks. In Stage 2, the input consisted of the predicted dose from Stage 1 concatenated with the geometric encoding maps ($M_{\text{sin}}, M_{\text{cos}}$) defined in Sec. 2.2.2, enabling direction-aware fluence prediction without redundant anatomical processing.

### 2.5. Experimental Setup

All experiments were implemented in `PyTorch` utilizing transformer backbones from the MONAI framework (Cardoso et al., 2022). The experiments were conducted on a Linux workstation using Python 3.10 and PyTorch 2.6 with CUDA 12.4 and cuDNN 9.1, on a

Table 1: Comprehensive comparison against naive baselines, strong literature baselines, and the proposed FluenceFormer framework. Best results are shown in bold.

| Category | Method/Backbone | MAE ↓ | Energy Err (%) ↓ | PSNR ↑ | SSIM ↑ |
|---|---|---|---|---|---|
| **Naive Baselines** *(Seg-Style)* | UNETR (Hatamizadeh et al., 2022) | $0.20 \pm 0.06$ | $22.1 \pm 8.4$ | $13.62 \pm 1.28$ | $0.43 \pm 0.06$ |
| | Swin UNETR (Hatamizadeh et al., 2021) | $0.11 \pm 0.01$ | $20.4 \pm 9.7$ | $15.53 \pm 1.48$ | $0.50 \pm 0.05$ |
| | nnFormer (Zhou et al., 2021) | $0.21 \pm 0.01$ | $22.1 \pm 9.3$ | $13.58 \pm 1.50$ | $0.45 \pm 0.07$ |
| | MedFormer (Gao et al., 2022) | $0.14 \pm 0.02$ | $20.3 \pm 8.4$ | $14.79 \pm 1.47$ | $0.40 \pm 0.03$ |
| **Strong Baselines** *(Direct Reg.)* | Standard CNN (U-Net) (Wang et al., 2020) | $0.05 \pm 0.06$ | $8.4 \pm 8.9$ | $16.00 \pm 1.57$ | $0.62 \pm 0.08$ |
| | Single-Stage Swin UNETR (Mgboh et al., 2025) | $0.05 \pm 0.07$ | $7.1 \pm 8.9$ | $16.45 \pm 1.46$ | $0.67 \pm 0.08$ |
| **FluenceFormer** *(Two-Stage)* | UNETR | $0.06 \pm 0.02$ | $7.0 \pm 9.6$ | $16.19 \pm 1.54$ | $0.60 \pm 0.08$ |
| | **Swin UNETR** | $\mathbf{0.02 \pm 0.06^{*}}$ | $\mathbf{6.1 \pm 8.9^{*}}$ | $\mathbf{17.99 \pm 1.59^{*}}$ | $\mathbf{0.70 \pm 0.07^{*}}$ |

*Indicates statistically significant improvement over Strong Baselines ($p < 0.05$).

single NVIDIA GeForce RTX 3090 GPU (24 GB). The dataset was partitioned into training (70%), validation (10%), and testing (20%) subsets. Models were optimized using the Adam optimizer ($lr=1 \times 10^{-4}$) with a batch size of 16 for 50 epochs per stage. While baseline models utilized a standard pixel-wise MSE loss, the FAR loss weights were set to (1, 0.5, 0.3, 0.2). Performance was evaluated using Mean Absolute Error (MAE), Peak Signal-to-Noise Ratio (PSNR), and Structural Similarity Index (SSIM). Crucially, to assess clinical deliverability, we report the Energy Error, calculated as the mean beam-wise relative deviation of the energy formulation defined in Eq. (2.3.1). This serves as a scale-invariant proxy for Monitor Unit (MU) accuracy. Statistical significance was verified using the Wilcoxon signed-rank test ($p < 0.05$).

## 3. Results

### 3.1. Comparative Analysis against State-of-the-Art Baselines

To validate the clinical efficacy of the proposed framework, we benchmarked **FluenceFormer** against two categories of existing methodologies: (1) *Naive Baselines*, utilizing sigmoid-based segmentation heads common in medical imaging; and (2) *Strong Baselines*, representing current state-of-the-art direct regression approaches, including a standard U-Net CNN and a Single-Stage Transformer. To isolate the architectural benefits of the proposed two-stage geometry-conditioned framework, the FluenceFormer models in this comparison are reported using standard MSE loss comparison before introducing the FAR loss. The single-stage Swin UNETR baseline in

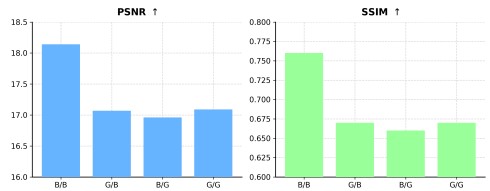

Figure 3: **Component-level ablation of FAR loss.** B/B yields the best PSNR and SSIM.

Table 2: **Backbone-Agnostic Validation of the FluenceFormer Framework.** We evaluate the proposed FAR loss against the MSE baseline across four distinct transformer architectures. The consistent improvement across all backbones demonstrates that the framework's efficacy is model-agnostic. Best results are shown in bold.

| Backbone | Loss Function | MAE ↓ | Energy Err (%) ↓ | PSNR ↑ | SSIM ↑ |
|---|---|---|---|---|---|
| **Swin UNETR** | Baseline (MSE) | $0.02 \pm 0.06$ | $6.10 \pm 8.90$ | $17.99 \pm 1.59$ | $0.70 \pm 0.07$ |
| | MSE + Energy | $0.03 \pm 0.01$ | $7.44 \pm 3.46$ | $17.12 \pm 1.52$ | $0.67 \pm 0.07$ |
| | MSE + Grad | $0.04 \pm 0.04$ | $6.82 \pm 2.07$ | $16.98 \pm 1.49$ | $0.67 \pm 0.08$ |
| | **Proposed FAR** | $\mathbf{0.02 \pm 0.01}$ | $\mathbf{4.53 \pm 2.54}^*$ | $\mathbf{18.14 \pm 1.57}$ | $\mathbf{0.76 \pm 0.08}^*$ |
| **UNETR** | Baseline (MSE) | $0.06 \pm 0.02$ | $7.01 \pm 9.60$ | $16.19 \pm 1.54$ | $0.60 \pm 0.08$ |
| | MSE + Energy | $0.05 \pm 0.01$ | $6.78 \pm 3.92$ | $17.19 \pm 1.36$ | $0.67 \pm 0.07$ |
| | MSE + Grad | $0.05 \pm 0.02$ | $7.08 \pm 2.75$ | $17.34 \pm 1.30$ | $0.52 \pm 0.08$ |
| | **Proposed FAR** | $\mathbf{0.04 \pm 0.01}$ | $\mathbf{6.74 \pm 2.15}^*$ | $\mathbf{17.21 \pm 1.42}$ | $\mathbf{0.67 \pm 0.07}^*$ |
| **nnFormer** | Baseline (MSE) | $0.03 \pm 0.01$ | $10.44 \pm 9.73$ | $17.73 \pm 1.41$ | $0.53 \pm 0.06$ |
| | MSE + Energy | $0.02 \pm 0.07$ | $11.87 \pm 3.28$ | $17.84 \pm 1.46$ | $0.56 \pm 0.06$ |
| | MSE + Grad | $0.03 \pm 0.01$ | $10.34 \pm 1.62$ | $17.80 \pm 1.45$ | $0.54 \pm 0.07$ |
| | **Proposed FAR** | $\mathbf{0.02 \pm 0.00}$ | $\mathbf{9.10 \pm 2.90}^*$ | $\mathbf{18.01 \pm 1.45}$ | $\mathbf{0.58 \pm 0.06}^*$ |
| **MedFormer** | Baseline (MSE) | $0.03 \pm 0.01$ | $8.79 \pm 7.16$ | $17.10 \pm 1.51$ | $0.60 \pm 0.06$ |
| | MSE + Energy | $0.03 \pm 0.02$ | $12.66 \pm 2.60$ | $17.71 \pm 1.47$ | $0.59 \pm 0.06$ |
| | MSE + Grad | $0.03 \pm 0.01$ | $10.72 \pm 2.04$ | $17.65 \pm 1.45$ | $0.59 \pm 0.06$ |
| | **Proposed FAR** | $\mathbf{0.02 \pm 0.07}$ | $\mathbf{7.50 \pm 2.20}^*$ | $\mathbf{17.85 \pm 1.50}$ | $\mathbf{0.65 \pm 0.06}^*$ |

$^*$Indicates statistically significant improvement over Baseline ($p < 0.05$).

Table 3: **Model Sensitivity Analysis.** Effect of input resolution, feature size, and output head on UNETR and Swin UNETR performance. The baseline configuration ($128 \times 128$, 48 feature size, Linear head) yields the best balance.

| Model | Image Size | Config | MAE ↓ | Energy Err (%) ↓ | PSNR ↑ | SSIM ↑ |
|---|---|---|---|---|---|---|
| **Swin UNETR** | $128 \times 128$ | 48, Linear | $\mathbf{0.02 \pm 0.01}$ | $\mathbf{4.53 \pm 2.54}$ | $\mathbf{18.14 \pm 1.57}$ | $\mathbf{0.76 \pm 0.08}$ |
| | $128 \times 128$ | 72, MLP | $0.03 \pm 0.01$ | $6.82 \pm 3.30$ | $17.15 \pm 1.46$ | $0.66 \pm 0.07$ |
| | $96 \times 96$ | 48, Linear | $0.02 \pm 0.01$ | $7.80 \pm 2.61$ | $17.05 \pm 1.46$ | $0.62 \pm 0.08$ |
| | $96 \times 96$ | 72, MLP | $0.03 \pm 0.01$ | $7.21 \pm 3.62$ | $17.10 \pm 1.45$ | $0.62 \pm 0.08$ |
| **UNETR** | $128 \times 128$ | 48, Linear | $\mathbf{0.04 \pm 0.01}$ | $\mathbf{6.74 \pm 2.15}$ | $\mathbf{17.21 \pm 1.42}$ | $\mathbf{0.67 \pm 0.07}$ |
| | $128 \times 128$ | 72, MLP | $0.04 \pm 0.01$ | $7.10 \pm 3.32$ | $17.15 \pm 1.45$ | $0.66 \pm 0.07$ |
| | $96 \times 96$ | 48, Linear | $0.05 \pm 0.01$ | $8.50 \pm 3.01$ | $16.50 \pm 1.45$ | $0.62 \pm 0.06$ |
| | $96 \times 96$ | 72, MLP | $0.05 \pm 0.02$ | $8.81 \pm 2.71$ | $16.10 \pm 1.39$ | $0.60 \pm 0.09$ |

Table 1 directly regresses fluence from CT and anatomical contours using a sigmoid output and MSE loss, without any intermediate dosimetric supervision. In contrast, FluenceFormer adopts an explicit two-stage formulation with intermediate dose prediction and geometry conditioned fluence regression using a ReLU-based physical regression head, such that the comparison contrasts learning formulations rather than constituting a strict architectural ablation.

Table 1 highlights the limitations of naive sigmoid baselines, which saturate at high intensities and yield $> 20\%$ Energy Errors. Strong regression baselines (ReLU) improve this to $\approx 7$–$8\%$, but the proposed **FluenceFormer** (Two-Stage Swin UNETR) achieves state-of-the-art performance. It reduces Energy Error to **6.1**% and yields a statistically significant improvement in SSIM (**0.70** vs. 0.67, $p < 0.05$), confirming that intermediate dose

Table 4: **Oracle-conditioned Stage 2 performance and slice-wise stability.** Stage 2 is conditioned on ground-truth dose to isolate the impact of the dose prior. Z-Consistency measures mean absolute differences between adjacent axial slices (lower is better).

| Model | MAE ↓ | PSNR ↑ | SSIM ↑ | Z-Consistency ↓ |
|---|---|---|---|---|
| Swin UNETR | $0.015 \pm 0.005$ | $20.5 \pm 2.1$ | $0.82 \pm 0.03$ | **0.10** |
| UNETR | $0.021 \pm 0.008$ | $19.8 \pm 2.5$ | $0.78 \pm 0.05$ | 0.26 |
| nnFormer | $0.019 \pm 0.006$ | $18.5 \pm 2.3$ | $0.60 \pm 0.04$ | 0.33 |
| MedFormer | $0.018 \pm 0.006$ | $18.8 \pm 2.2$ | $0.71 \pm 0.04$ | 0.25 |

supervision better resolves fine-grained IMRT modulation. Figure 2 visually corroborates these findings across representative projection angles (Beams 1, 4, and 5), which exhibit complex internal intensity modulation. While the **UNETR (Seg-style)** baseline fails to capture this modulation due to saturation artifacts, and the **Standard CNN** suffers from blurring despite capturing general intensity, **FluenceFormer** (SwinUNETR) accurately reconstructs the full physical intensity spectrum. The proposed framework faithfully preserves both sharp field edges and fine-grained internal modulation, validating the benefits of physical regression in suppressing artifacts without sacrificing peak accuracy. In order to quantify the effect of Stage 1 inaccuracies on Stage 2 fluence prediction, we additionally evaluate an oracle setting in which Stage 2 is conditioned on ground-truth dose rather than the predicted dose. This analysis isolates the impact of the dose prior while keeping the Stage 2 architecture and FAR loss unchanged. Table 4 reports the resulting oracle-conditioned Stage 2 performance across all backbones.

### 3.2. Strict Single-Stage Ablation

As shown in Table 5, removing the intermediate dose prediction stage consistently degrades fluence accuracy and energy consistency across all backbones. Although the single-stage Swin UNETR is the strongest direct-regression baseline (MAE = 0.04, Energy Error = 7.54%), its error remains substantially higher than the corresponding two-stage Fluence-Former variant. This indicates that the gains of FluenceFormer arise from explicit dose-conditioned decomposition rather than backbone capacity or geometry encoding alone.

### 3.3. Backbone-Agnostic Validation

To demonstrate that **FluenceFormer** is a generalized framework rather than a model-specific improvement, we evaluated the proposed methodology across four distinct transformer architectures: Swin UNETR, UNETR, nnFormer, and MedFormer. Table 2 details the perfor-

Table 5: **Strict single-stage ablation study.** All models directly regress fluence maps from CT, contours, and beam geometry in a single stage using identical backbones, regression heads, and training settings.

| Model | MAE ↓ | Energy Error (%) ↓ |
|---|---|---|
| UNETR | $0.06 \pm 0.03$ | $9.31 \pm 6.78$ |
| Swin UNETR | $\mathbf{0.04 \pm 0.01}$ | $\mathbf{7.54 \pm 7.30}$ |
| nnFormer | $0.07 \pm 0.02$ | $10.12 \pm 13.87$ |
| MedFormer | $0.07 \pm 0.03$ | $10.44 \pm 14.08$ |

mance of each backbone when trained with the standard Baseline (MSE) versus the proposed

FAR loss. We also include some comparison against different loss function combinations such as MSE + Energy, MSE + Grad, to isolate the contributions of specific physical constraints. Across all architectures, the FAR formulation yields consistent, statistically significant improvements ($p < 0.05$) over MSE baselines. For instance, Swin UNETR reduces Energy Error from 6.10% to 4.53%, while nnFormer improves structural fidelity. Crucially, every backbone utilizing the FAR loss outperforms the strong external baselines reported in Table 2, confirming that the FAR loss is model-agnostic. This demonstrates that the superior clinical efficacy is driven by the unified physics-aware methodology rather than a dependency on a specific architecture.

### 3.4. Model Sensitivity Analysis

We evaluated the framework's robustness against variations in input resolution, feature dimension, and decoder complexity (Table 3). The baseline configuration (128 × 128, feature size 48, Linear head) consistently yielded the optimal balance of structural fidelity and phys-

Table 6: Ablation on correlation and energy computation scopes within the FAR loss. All experiments use $\alpha/\beta/\gamma/\delta = 1 / 0.5 / 0.3 / 0.2$. Results are reported as Mean ± Standard Deviation. The fully Beam-wise configuration (B/B) yields the highest structural fidelity.

| ID | Corr Scope | Energy Scope | PSNR ↑ | SSIM ↑ |
|---|---|---|---|---|
| **B/B** | Beam-wise | Beam-wise | $\mathbf{18.14 \pm 1.57}$ | $\mathbf{0.76 \pm 0.08}$ |
| G/B | Global | Beam-wise | $17.07 \pm 1.48$ | $0.67 \pm 0.07$ |
| B/G | Beam-wise | Global | $16.96 \pm 1.56$ | $0.66 \pm 0.07$ |
| G/G | Global | Global | $17.09 \pm 1.50$ | $0.67 \pm 0.07$ |

ical accuracy. For Swin UNETR, reducing spatial resolution to 96 × 96 increased Energy Error from 4.53% to 7.80%, indicating that hierarchical self-attention requires fine-grained tokens to resolve steep dose gradients. Similarly, replacing the linear head with a deeper MLP degraded SSIM ($0.76 \rightarrow 0.66$), suggesting that non-linear projections induce regression instability.

#### 3.4.1. Local vs. Global Loss Formulation

We investigated the impact of spatial scope on the physics-informed constraints by computing correlation and energy terms either locally per-beam (**B**) or globally across the batch (**G**). As shown in Table 6 and Figure 3, the fully beam-wise formulation (**B/B**) yields superior performance, achieving a PSNR of $\mathbf{18.14}$ dB and SSIM of $\mathbf{0.76}$. Notably, introducing global aggregation in either term degrades structural fidelity, causing an SSIM drop of approximately 0.09. This confirms that enforcing physical constraints independently per beam is essential to preserve distinct IMRT modulation patterns, whereas global averaging dilutes beam-specific gradients.

### 3.5. Clinical Dose Evaluation

To assess the clinical feasibility of the predicted fluence maps, we forward-calculated dose distributions in the Eclipse TPS using the predicted fluences and identical beam geometries and machine parameters as the reference clinical plans.

Figure 4 shows representative DVHs for target volumes and organs-at-risk, comparing recalculated doses from FluenceFormer variants with clinical plans. Across evaluated cases,

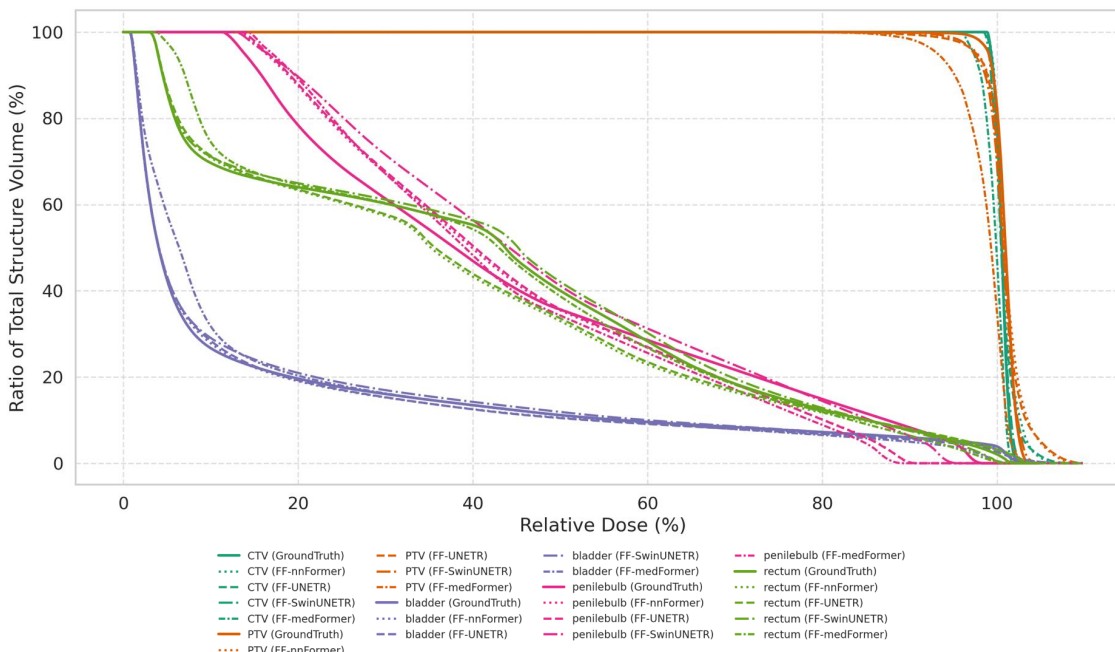

Figure 4: **DVH comparison using TPS-recalculated dose from predicted fluences.**

| Backbone | Trainable Params (M) | Inference Time (s) |
|---|---|---|
| SwinUNETR | 25.14 | 0.55 |
| UNETR | 91.47 | 0.33 |
| nnFormer | 23.25 | 0.36 |
| MedFormer | 10.88 | 0.34 |

Table 7: Trainable parameters and inference time per patient.

DVHs closely match the reference, indicating preserved target coverage and organ-at-risk sparing. Deliverability was further assessed via 3D gamma analysis ($3\%/3\,\mathrm{mm}$). Across test cases, Swin UNETR achieves the highest pass rate ($92 \pm 8.01\%$), followed by UNETR ($90 \pm 10.67\%$), MedFormer ($87 \pm 8.71\%$), and nnFormer ($85 \pm 7.65\%$).

### 3.6. Runtime and Resource Efficiency

End-to-end inference, including Stage 1 dose prediction and Stage 2 fluence regression, requires $0.55\,\mathrm{s}$ per patient for Swin UNETR and $0.33$–$0.55\,\mathrm{s}$ across all backbones (Table 7), with peak GPU memory usage of only $3.2\%$ of a $24\,\mathrm{GB}$ GPU, indicating computational efficiency suitable for clinical deployment.

## 4. Discussion

This study establishes **FluenceFormer** as a solution to the ill-posed mapping from anatomy to machine parameters. Our results identify the intermediate dose distribution as a critical "structural bridge," translating anatomical constraints into a realizable target before beam modulation. This challenges direct end-to-end regression, as without this dosimetric prior,

models struggle to resolve the spatial ambiguity of overlapping beams. Furthermore, sinusoidal encoding shows that distinguishing directional intensity does not require expensive convolutions, but benefits from explicit geometric conditioning.

Crucially, we validate the necessity of continuous physical regression over segmentation-style formulations. While sigmoid-based baselines saturate at high intensities, yielding Energy Errors $> 20\%$, FluenceFormer's ReLU activation enables the unbounded regression of physical Monitor Units. When regularized by the **Fluence-Aware Regression (FAR)** loss, specifically the beam-wise energy term, this formulation reduces Energy Error to $\approx 6\%$ and achieves statistically significant structural gains over strong CNN baselines (SSIM 0.70 vs. 0.67, $p < 0.05$).

Beyond fluence-domain accuracy, closed-loop dose evaluation substantiates the clinical relevance of FluenceFormer. Forward-calculated doses from predicted fluences closely match reference plans, with overlapping DVHs for targets and organs-at-risk. 3D gamma analysis ($3\%/3\,\mathrm{mm}$) yields high pass rates across all variants (up to $92 \pm 8.01\%$ for Swin UNETR), confirming physically deliverable and dosimetrically consistent treatment plans and demonstrating that the two-stage formulation preserves clinical dose quality beyond fluence similarity.

Architecturally, **Swin UNETR** proved superior, as its hierarchical window-based attention strikes the optimal balance between global reasoning and local spatial preservation. In contrast, **UNETR**'s reliance on "flat token" sequence modeling limits its ability to resolve high-frequency spatial details, resulting in the blurred edges observed in our qualitative analysis. Similarly, while **MedFormer** (B-MHA) and **nnFormer** (interleaved attention) offer strong segmentation capabilities, they appear less effective at regressing the continuous, unbounded intensity values required for physical fluence. This suggests that Swin UNETR's shifted-window mechanism is uniquely suited for the dense, gradient-heavy nature of IMRT modulation.

From a learning perspective, the primary limitation is the potential bias toward the specific distributional priors of a single dataset, which may restrict generalization to unseen clinical settings. While our geometric encoding is theoretically agnostic, its ability to generalize to diverse beam configurations and planning strategies remains empirically unverified. Furthermore, the scarcity of standardized open-source benchmarks for fluence prediction hinders rigorous reproducibility and fair comparison across methods. Future work will extend the framework to additional treatment sites and variable beam configurations, as well as investigate domain adaptation techniques to ensure robustness across heterogeneous imaging protocols. In addition, we plan to integrate differentiable dose calculation layers to fully close the loop between predicted fluence and the resulting delivered dose, enabling end-to-end physically consistent optimization.

In conclusion, **FluenceFormer** establishes a robust and effective baseline for transformer-based radiotherapy planning. By enforcing physical consistency through the FAR loss and resolving geometric ambiguity via explicit conditioning, it offers a data-efficient and interpretable pathway toward automated IMRT planning. More broadly, this work highlights the value of incorporating physics-aware constraints and geometric conditioning into transformer architectures for accurate and reliable fluence map prediction within radiotherapy planning. These findings demonstrate the strong potential of combining domain knowledge with deep learning techniques to achieve clinically meaningful treatment planning solutions.

## Acknowledgments

The authors gratefully acknowledge the support of the Thomas Rumble Fellowship from Wayne State University and Henry Ford Health, which made this research possible.

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

## Appendix A. Additional Figure(s)

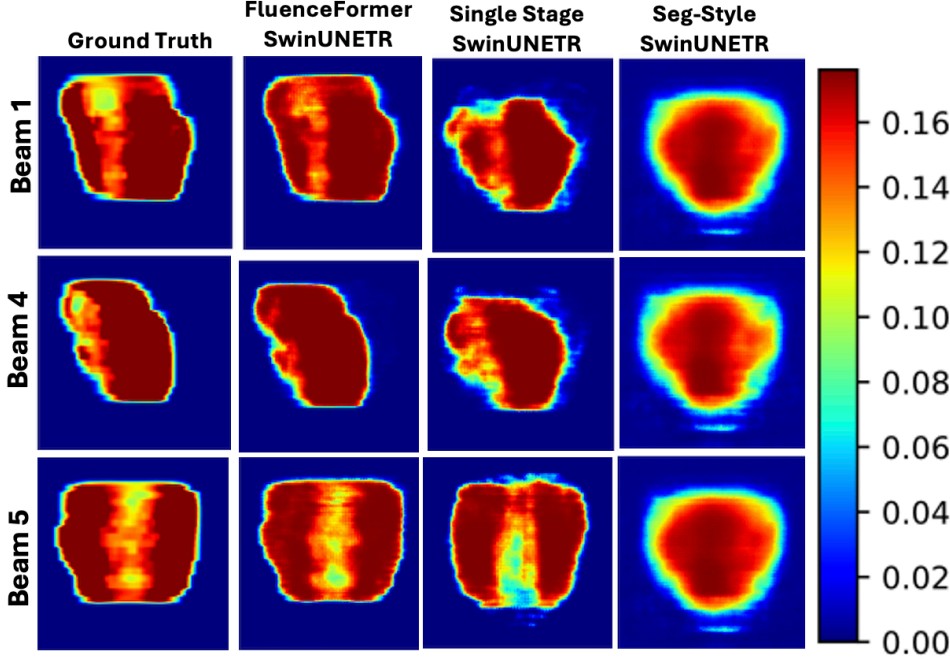

Figure 5: **Controlled qualitative comparison with a fixed backbone.** Representative beam-wise fluence maps comparing (from left to right): ground truth, two-stage Fluence-Former, single-stage regression, and segmentation-style prediction, *all implemented using the same Swin UNETR backbone.* Differences therefore arise solely from the learning formulation rather than backbone capacity, addressing the fairness concern of Fig. 2 in the main paper.

