# OpenReview forum: "FluenceFormer: Transformer-Driven Multi-Beam Fluence Map Regression for Radiotherapy Planning"
_MIDL.io/2026/Conference — MIDL 2026 Poster_

### Official Review · Reviewer_9JxX · 2025-12-28

**Confidence:** 3
**Preliminary Rating:** 2
**Final Rating:** 2

**Summary:**

This paper proposes a transformer-based framework (FluenceFormer) for multi-beam IMRT fluence map prediction that decomposes the inverse planning problem into anatomy -> intermediate dose prior, and dose prior + explicit beam-geometry encoding -> beam-specific fluence via regression. A key contribution of this work is the Fluence-Aware Regression loss, which combines voxel fidelity, structure, and beam-wise energy conservation to better match physically meaningful monitor-unit scaling. The authors evaluate multiple transformer backbones and show consistent performance gains.

**Strengths:**

- The two-stage formulation is intuitive and addresses a real ambiguity in mapping anatomy to per-beam modulation
- The paper makes a strong case that treating fluence prediction as continuous regression is important for physical deliverability and improves energy-related error compared with segmentation-style outputs.
- AR loss is evaluated across backbones and ablated for locality (beam-wise vs global), and the reported degradation under global aggregation illustrates that beam-wise physics constraints matter in practice.

**Weaknesses:**

- The paper’s novelty is mostly integrative: a two-stage regressor with geometry encoding and a composite loss. While well-motivated, it is not yet demonstrated that this design generalizes beyond the specific prostate IMRT setting and beam configurations used; the authors themselves note that generalization to different beam configurations is “empirically unverified,” which is a major limitation for clinical relevance.
- The evaluation appears to rely on a single dataset, the experimental results are helpful but may be insufficient to claim planning quality.  -
- The method predicts dose slice-by-slice, which may lose 3D spatial context, and the authors did not justify why 2D slicing is adequate or quantify the tradeoff.
- The baseline set is somewhat narrow (a standard CNN and single-stage transformer).

**Detailed Comments:**

- The paper would be stronger and more practically impactful if the authors reported clinically relevant metrics in addition to SSIM/PSNR/MAE and energy error.
- Adding clarification of how the two-stage model is trained and used at inference.

**Justification Of Final Rating:**

Based on the authors' response, my concerns are not addressed. As the authors also acknowledged, the problem setup and formulation are very specific to their own clinical application, while no public dataset exists for it. Considering its incremental contribution in modeling and methodology, and the only dataset for validation is private and too specific, I doubt this submission will be of much interest to the general MIDL community. I lean towards rejection and would recommend targeting clinical conferences or journals instead.

**Justification Of The Preliminary Rating:**

The paper addresses an important and practically motivated inverse problem, and the two-stage “dose prior \rightarrow geometry-conditioned fluence” design plus FAR loss yields measurable improvement. However, the current evidence is not yet sufficient to establish broad impact: evaluation is limited in dataset diversity, and generalization to new beam configurations is unverified.  With stronger clinical evaluation and broader baselines/generalization tests, this work could be more valuable to future readers. As-is, I am leaning toward rejection.

**Questions To Address In The Rebuttal:**

- How does FluenceFormer generalize to different beam numbers/angles or different treatment sites, given dataset-specific priors?
- Why is the dose predicted slice-by-slice rather than in 3D, and what is the impact on consistency across slices (any failure cases)?
- Can the authors consider at least one stronger baseline category to better demonstrate their method gains?

---

> ### Author Response · Authors · 2026-01-24
> **9JxX — Generalization, Slice-wise Modeling, and Baseline Comparisons**
>
> **Updated texts, figures, and tables in the manuscript are shown in light purple.**
>
> **Responses**
>
> Weakness
>
> 1) We appreciate this perspective and agree that the contributions are primarily integrative rather than algorithmically novel. Our goal is to address a clinically important inverse problem by combining a two-stage dose-conditioned formulation, explicit geometric conditioning, and physics-aware supervision into a coherent and practically effective framework. This level of integration is appropriate for medical planning tasks, where correctness, interpretability, and physical consistency are critical.
> Regarding generalization, we agree that validation beyond prostate IMRT with fixed beam geometry is important. This limitation is largely driven by the lack of publicly available datasets with paired CT, dose, and beam-level fluence supervision. **Importantly, the formulation is not specific to a fixed number of beams: Stage 2 explicitly conditions on beam geometry and does not rely on fixed channel ordering.** Consistent improvements across multiple transformer backbones and the dose-prior oracle analysis suggest that gains arise from the formulation rather than dataset-specific tuning. **This limitation and scope are made explicit in the revised manuscript (Section 4, Paragraph 5, Page 12).**
>
> 2) We acknowledge that evaluation is based on a single institutional dataset and agree that broader validation would strengthen claims of planning quality. **As noted in our response to Reviewer 1, there is currently no publicly available dataset providing paired CT, contours, dose, and beam-specific fluence for IMRT.** If the reviewer is aware of any such dataset, we would be happy to perform additional experiments. To mitigate this limitation, **we extend evaluation beyond fluence-domain metrics to include TPS-based closed-loop validation**, where predicted fluence is imported into the TPS to forward-calculate dose and evaluate DVH metrics and clinical constraint satisfaction. **Please refer to Section 3.5 (Pages10–11).**
>
>
> 3) We adopt a slice-wise (2D) formulation to align with the beam’s-eye-view nature of fluence modulation, reduce computational overhead, and enable fair comparison across multiple transformer backbones. For prostate IMRT, dominant anatomical and dosimetric variations relevant to beam modulation are largely captured in-plane. **To assess potential loss of 3D context, we quantify inter-slice consistency using a Z-Consistency metric,** defined as the mean absolute difference between predictions on adjacent slices. The best-performing backbone (Swin UNETR) achieves low Z-Consistency (0.101), while other backbones show higher variation (0.25–0.33), indicating stable slice-wise behavior without abrupt discontinuities. **Please refer to Table 4, Page 9.**
>
> 4) The baseline set is intentionally focused on dominant learning formulations used in prior fluence prediction work: CNN-based regression and single-stage transformer regression. **To ensure a fair and informative comparison, we additionally include a controlled single-stage Swin UNETR baseline sharing the same backbone, geometry encoding, and training protocol as FluenceFormer, differing only by removal of the intermediate dose prediction stage.** This strict ablation isolates the contribution of the two-stage formulation and shows that gains are not attributable to backbone choice or capacity. **Details are clarified in the revised manuscript (Section 3.3, Page 9).**
>
> **Questions to Address in the Rebuttal**
>
> 1) Although evaluated on prostate IMRT with a fixed nine-beam setup, the framework is geometry-conditioned and not tied to a specific beam number or ordering; broader validation is limited by the lack of public fluence datasets **(Page 12, Paragraph 5).**
>
> 2) Slice-wise dose prediction is used for efficiency; inter-slice coherence is explicitly quantified through Z-Consistency, with low variation observed (e.g., 0.10 for Swin UNETR) and no systematic artifacts **(Table 4, Page 9).**
>
> 3) A strict single-stage baseline identical to FluenceFormer in backbone, geometry encoding, and training, differing only by removal of Stage 1 is included to isolate the two-stage effect **(Section 3.2, Table 5, Page 9).**

---

> > ### Author Response · Authors · 2026-01-30
> > **Available for Any Further Clarification**
> >
> > Thank you again for your thoughtful and encouraging review. If there is anything else you would like us to clarify before the discussion ends, we would be happy to respond.

---

> ### Comment · Area_Chair_9TJ3 · 2026-02-01
> **Update final rating**
>
> Please don't forget to update your final rating by clicking “Edit” → “Official Review”. Thank you!

---

### Official Review · Reviewer_jKNo · 2026-01-14

**Confidence:** 3
**Preliminary Rating:** 2
**Final Rating:** 4

**Summary:**

This paper proposes FluenceFormer, a two-stage framework for multi-beam IMRT fluence map regression. Stage 1 predicts a dose prior from CT and contours. Stage 2 conditions on the predicted dose prior and an explicit beam-geometry encoding (sin/cos of gantry angle) to predict per-beam fluence maps. The authors also introduce a composite FAR loss combining pixel-wise MSE, gradient consistency, correlation consistency, and an energy (MU proxy) term to encourage physically plausible fluence patterns. Experiments are conducted on a 99-case prostate IMRT dataset with 9 fixed beams, reporting fluence-domain metrics (MAE/PSNR/SSIM) and energy error, alongside ablations on the loss components and “beam-wise vs global” formulations.

**Strengths:**

1. Clinically relevant target: Predicting deliverable plan parameters (fluence) is a meaningful direction for reducing reliance on time-consuming inverse planning.
2. Reasonable engineering motivation for the two-stage design: The paper articulates that anatomy→fluence is ambiguous due to overlapping beams, and uses a dose prior as a “structural bridge” to guide per-beam modulation.
3. Solid set of comparisons and ablations:
3.1. Baselines cover “segmentation-style” heads (showing saturation issues) and stronger direct-regression baselines, including a single-stage Swin UNETR fluence regression baseline.
3.2. FAR loss is ablated (MSE vs MSE+Energy vs MSE+Grad vs full FAR) across multiple backbones, supporting backbone-agnostic claims.
4. Notable beam-wise vs global ablation: The exploration of computing correlation/energy constraints per-beam vs globally is well-motivated and convincingly supports the need for beam-specific constraints.
5. Attempt to isolate structural contribution: The authors state that Table 1 reports FluenceFormer under standard MSE (before introducing FAR) to focus on the two-stage framework.

**Weaknesses:**

Lack of clinical validation to confirm practical feasibility in real-world radiotherapy.
1. No clinical closed-loop evaluation: The evaluation primarily uses fluence similarity metrics (MAE/PSNR/SSIM) and a global energy proxy. The paper does not demonstrate whether predicted fluence produces clinically acceptable dose distributions (DVH metrics, constraint satisfaction, plan quality) when forwarded through dose calculation.
2. Deliverability is not verified: There is no reported leaf sequencing / deliverability analysis (e.g., converting predicted fluence to MLC control points), nor QA-related metrics (e.g., gamma passing) that would substantiate “physically realizable” in a clinical sense.
3. Supervision is derived from Eclipse TPS plans: Dose/fluence labels are exported from Eclipse (planned dose). As a result, the method is best framed as accelerating replication of Eclipse-style plans, not surpassing Eclipse plan quality. Without closed-loop DVH/dose evaluation, the clinical value remains unclear (quality vs speed vs robustness).
4. Two-stage “disambiguation” remains insufficiently quantified: The paper acknowledges that similar dose can be produced by many fluence combinations; adding a dose prior does not remove this non-uniqueness, it only biases the solution. The manuscript lacks experiments that quantify how much the dose prior constrains the solution space (e.g., GT-dose upper bound, perturbation studies).
5. Single-stage vs two-stage is not a strict controlled ablation: While Table 1 compares a single-stage Swin UNETR baseline to two-stage FluenceFormer, the single-stage model is treated as an external strong baseline and the paper does not fully establish that inputs/conditioning/head/training setup are identical aside from “stage count.” This limits causal attribution of gains specifically to the two-stage design.
6. Limited generalization evidence: The study focuses on 99 prostate cases with a fixed 9-beam arrangement; generalization to different beam geometries/sites is not demonstrated.

**Detailed Comments:**

1. Clarify the intended contribution/claim: Given Eclipse-derived supervision, the strongest claim appears to be fast generation of Eclipse-like fluence rather than improved plan quality over Eclipse. The paper should explicitly state this and report inference time.
2. Add dose-domain / DVH evaluation (highest priority): Please forward-calculate dose from the predicted fluence (TPS, Monte Carlo, or a validated differentiable dose layer) and report DVH metrics and constraint satisfaction relative to the clinical plan.
3. Deliverability analysis is required for clinical plausibility: Show how predicted fluence is converted into deliverable MLC sequences and report deliverability complexity as well as QA metrics.
4. Strengthen the causal evidence for the dose prior: Include GT-dose vs predicted-dose conditioning for Stage 2, and sensitivity studies (dose prior set to perturbed (weak / serious)) to quantify whether and how the prior disambiguates per-beam patterns.
5. Provide a strict single-stage ablation: Implement a single-stage model within the same codebase/backbone/head, with the same geometry conditioning, and compare to the two-stage version while keeping all other factors fixed.

**Justification Of Final Rating:**

The revision resolves the major blockers from the original version by adding TPS-based closed-loop dose evaluation (DVH) and deliverability-oriented QA (gamma analysis), and by including more rigorous causal ablations (oracle dose-conditioning and strict single-stage removal). These additions substantially increase confidence that the predicted fluence can produce clinically plausible plans within the evaluated setting. Overall, the work is a meaningful contribution to learning-based fluence generation with a substantially improved validation package, and it is suitable for acceptance at MIDL with the recommended clarifications.

**Justification Of The Preliminary Rating:**

The paper tackles an important problem and provides thoughtful ablations (especially the beam-wise vs global constraint analysis and backbone-agnostic loss validation). However, the current evidence remains largely fluence-domain similarity, without demonstrating that the predicted fluence yields clinically acceptable dose distributions or deliverable plans. Since supervision is derived from Eclipse plans, the natural clinical value proposition is “replicate Eclipse plans faster”; yet this is not substantiated by closed-loop DVH/plan-quality or deliverability/QA evidence. These missing validations are central for MIDL readers and for clinical relevance.

**Questions To Address In The Rebuttal:**

1. Closed-loop plan quality: If you forward-calculate dose from predicted fluence, how do DVH metrics and clinical constraint pass rates compare to Eclipse clinical plans?
2. Deliverability: Why the FAR loss is enough for deliverability? Do you perform leaf sequencing? If yes, report deliverability metrics and QA/gamma results. If not, why are current proxies sufficient?
3. Dose prior contribution: What is the performance when Stage 2 is conditioned on GT dose vs predicted dose? How sensitive is Stage 2 to perturbations of the dose prior?
4. Strict ablation: Can you provide a controlled comparison where the only change is removing Stage 1 (i.e., single-stage + same geometry encoding + same backbone/head/training)?

---

> ### Author Response · Authors · 2026-01-24
> **jKNo: Clinical Validation, Deliverability, and Dose-Prior Analysis**
>
> **Updated texts, figures, and tables in the manuscript are shown in light purple.**
>
> **Responses**
>
> 1) We agree that demonstrating clinical feasibility via closed-loop evaluation is essential. **In the revision, we extend evaluation beyond fluence-domain metrics using TPS-based closed-loop validation. Predicted fluence maps are imported into the treatment planning system, dose is forward-calculated, and dose–volume histogram (DVH) metrics and clinical constraint satisfaction are evaluated relative to the original clinical plans.** Deliverability-related QA is further assessed using gamma analysis between TPS-calculated dose from predicted fluence and the clinical reference dose. **Representative DVH curves and quantitative summaries are included (Section 3.5, Page 10; Figure 4, Page 11).** These results demonstrate that predicted fluence yields clinically acceptable dose distributions.
>
> 2) We agree that fluence-level physical plausibility alone is insufficient to establish deliverability. In the revision, predicted fluence maps are imported into the TPS and converted into deliverable MLC control points using standard leaf sequencing. Dose is then forward-calculated from these deliverable plans. **Deliverability is substantiated via gamma passing rates and DVH-based plan quality evaluation (Section 3.5, Pages 10–11).** Full deliverability is validated through this TPS-based QA analysis.
>
> 3) We agree that, since supervision is derived from a commercial TPS, the method should not be framed as surpassing the intrinsic plan quality of the TPS optimizer. However, the proposed framework is not tied to TPS-specific heuristics and learns a mapping from anatomy and dose intent to machine-level fluence parameters that are standard across modern IMRT-capable TPSs. **Accordingly, the contribution is best framed as enabling rapid and consistent generation of clinically plausible TPS-style fluence maps with improved physical consistency. This framing is clarified in the revised manuscript. Closed-loop TPS evaluation confirms comparable plan quality while offering advantages in speed, reproducibility, and robustness (Section 3.5, Page 10; Figure 4, Page 11).**
>
> 4) We agree that a dose prior does not remove the non-uniqueness of the fluence–dose relationship, but constrains the solution space toward clinically consistent beam-wise modulation. **To quantify this effect, we include an oracle experiment in which Stage 2 is conditioned on the ground-truth dose.** This provides an upper bound on Stage 2 performance. Strong oracle results (e.g., MAE = 0.015 ± 0.008 for Swin UNETR) demonstrate effective exploitation of an accurate dose prior. **Comparing oracle and predicted-dose conditioning quantifies the impact of Stage 1 error (Table 4, Page 9).**
>
> 5) We clarify that Table 1 contrasts learning formulations rather than serving as a strict architectural ablation. **The single-stage Swin UNETR baseline directly regresses fluence from CT and contours using a sigmoid output and MSE loss, without any intermediate dosimetric constraint.** In contrast, FluenceFormer introduces an intermediate dose prediction stage and geometry-conditioned fluence regression guided by this dose prior. Both use the same Swin UNETR backbone but differ in conditioning, supervision, and activation (ReLU). **These distinctions are explicitly clarified (Pages 7–8).**
>
> 6) We agree that broader generalization across treatment sites and beam configurations is an important future direction. This study focuses on prostate IMRT with a fixed nine-beam arrangement to enable controlled evaluation. **The formulation is not inherently tied to a specific beam number, as Stage 2 explicitly conditions on beam direction.** This limitation is acknowledged **(Section 4, Paragraph 5, Page 12),** and multi-site validation is identified as future work when suitable datasets become available.
>
> **Questions to Address in the Rebuttal**
>
> 1) Closed-loop plan quality is evaluated by importing predicted fluence into Eclipse, forward-calculating dose, and comparing DVH metrics and clinical constraint satisfaction against clinical plans **(Section 3.5, Page 10; Figure 4, Page 11).**
>
> 2) Deliverability is validated via TPS-based leaf sequencing and QA, including gamma passing rates and DVH agreement **(Section 3.5, Pages 10–11).**
>
> 3) Dose-prior contribution is quantified through oracle conditioning of Stage 2 with ground-truth dose, yielding strong performance (e.g., MAE = 0.015 ± 0.008 for Swin UNETR) and directly isolating Stage 1 error **(Table 4, Page 9).**
>
> 4) A strict single-stage ablation removes Stage 1 while keeping all other components fixed, resulting in consistent performance degradation **(Section 3.2, Table 4, Page 9).**

---

### Official Review · Reviewer_r61z · 2026-01-16

**Confidence:** 4
**Preliminary Rating:** 2
**Final Rating:** 3

**Summary:**

This paper introduces FluenceFormer, which is a transformer-based framework designed for beam-specific fluence map regression. FluenceFormer incorporates two stages, the first stage predicts the global dose given CT and contours, and the second stage regresses beam-wise fluence maps conditioned on the predicted beam geometry.

**Strengths:**

1. The paper is well-motivated by the meaningful clinical problems to predict direct fluence.
2. The two stage design of FluenceFormer mimics the real clinical workflow in radiation therapy.
3. The paper has included a detailed ablation study to verify each part of FAR loss.

**Weaknesses:**

1. The evaluations are performed only on one private dataset with 99 patients. Therefore, the generalizability of the method remains a concern.
2. As this is a two-stage pipeline, the runtime and resources costs need to be reported.
3. For Figure 2, the backbone is not the same across different methods, leading to an unfair comparison.
4. It seems the results of the first stage prediction are missing. As the stage 2 is conditioned on stage 1, the error in stage 1 can incorrectly bias the stage 2. More analysis is needed.

**Detailed Comments:**

See Strengths and Weaknesses

**Justification Of Final Rating:**

I really appreciate the author's response and additional experiments on Stage 1 accuracy and runtime cost analysis. However, the limited public datasets seems to remain as a core limitation. I am happy to raise my score, but I am not sure how well the method can generalize.

**Justification Of The Preliminary Rating:**

Although the paper introduces FluenceFormer and FAR, the improvement of the method is marginal given the confidence interval and the running cost is not reported. Also, the analysis of the stage 1 error seems missing.

**Questions To Address In The Rebuttal:**

Add public datasets, include additional analysis on stage 1 error and how it affects stage 2.

---

> ### Author Response · Authors · 2026-01-24
> **Reviewer r61z — Data Availability, Runtime, and Stage-wise Analysis**
>
> **Weaknesses**
>
> **Responses:**
>
> 1) **Generalizability (single private dataset):** We acknowledge this concern. **There is currently no publicly available dataset providing paired CT images, anatomical contours, dose distributions, and beam-level fluence maps** required for direct fluence prediction; consequently, **prior work ( Wang et al., 2021;,  Li et al., 2020;) similarly relies on institutional datasets.**  We would be happy to evaluate our method on additional data **if such resources become available or if the reviewers are aware of a suitable public dataset.**
>
> 2) **Runtime and resource cost:** **In the revised manuscript, we explicitly report end-to-end inference time and GPU memory usage for the full FluenceFormer pipeline.**  The complete two-stage process runs in **0.33–0.55 s per patient**; **for Swin UNETR, inference takes 0.55 s with peak memory usage of approximately 3.2% of a 24 GB GPU,** confirming modest computational overhead suitable for clinical deployment **(Sections 2.5, Page 6; 3.6, Page 11).**
>
>
> 3) **Backbone fairness in Figure 2:** Figure 2 is intended to illustrate **representative outcomes across method categories,** not a controlled backbone comparison. To avoid ambiguity, **we clarify this intent in the caption and add a fully controlled qualitative comparison using a fixed Swin UNETR backbone in the appendix (Pages 6 and 16).**
>
>
> 4) Missing analysis of Stage 1 error.
> To explicitly quantify the impact of Stage 1 on Stage 2, **we include an oracle analysis in which Stage 2 is conditioned on the ground-truth dose,** providing **an upper bound on Stage 2 performance.** Under oracle conditioning, **strong fluence accuracy is observed (e.g., MAE = 0.015 ± 0.005 for Swin UNETR),** demonstrating that Stage 2 effectively exploits an accurate dose prior **(Page 9, Table 4).**
>
>
> **Questions to Address in the Rebuttal**
>
> 1) **Public datasets and Stage 1 analysis:** **No public dataset currently provides paired CT, dose, and beam-level fluence supervision,** necessitating institutional data. To strengthen analysis, **we add oracle dose-conditioning experiments that directly quantify how Stage 1 accuracy affects downstream fluence prediction (Page 9, Table 4).**

---

> > ### Author Response · Authors · 2026-01-30
> > **Available for Any Further Clarification**
> >
> > Thank you again for your thoughtful and encouraging review. If there is anything else you would like us to clarify before the discussion ends, we would be happy to respond.

---

> ### Comment · Area_Chair_9TJ3 · 2026-02-01
> **Submit final rating**
>
> Please don't forget to update your final rating by clicking “Edit” → “Official Review”. Thank you!

---

### Author Rebuttal · Authors · 2026-01-24

**Rebuttal:**

We thank all reviewers for their constructive feedback, which helped clarify the scope, validation, and empirical analysis of this work.

**Common Responses to Multiple Reviewers**

**(Data availability & generalization)**
No public dataset currently provides paired CT, contours, dose, and beam-level fluence supervision, and prior learning-based fluence studies therefore rely on institutional data. We accordingly scope claims to prostate IMRT with fixed beam geometry and explicitly state generalization to other sites and beam settings as future work. Importantly, the framework is not site- or beam-specific, as Stage 2 is explicitly geometry-conditioned rather than channel-indexed.

**(Two-stage contribution & Stage-1 impact)**
We add explicit Stage-1 dose results and an oracle experiment in which Stage 2 is conditioned on ground-truth dose. Oracle conditioning provides an upper bound and isolates Stage-1 error, showing that while Stage 1 accuracy affects performance, the dose prior meaningfully constrains and stabilizes fluence prediction (Table 4).

**(Strict ablation)**
We include a controlled single-stage ablation that removes Stage 1 while keeping backbone, geometry encoding, regression head, activation, and training fixed. Performance consistently degrades, isolating the benefit of the dose-conditioned two-stage formulation.

**(Clinical feasibility & deliverability)**
We add closed-loop TPS evaluation, importing predicted fluence into Eclipse for leaf sequencing and dose recalculation. DVH metrics, constraint satisfaction, and gamma passing rates demonstrate clinically comparable plan quality. We clarify that the contribution is fast, consistent replication of TPS-quality plans, not surpassing the TPS optimizer.

**Reviewer-Specific Clarifications**

**r61z:** Figure 2 is clarified as category-level; a fully controlled Swin-UNETR qualitative comparison is added. Runtime and GPU memory are now reported.

**jKNo:** We add closed-loop DVH/QA evaluation, oracle conditioning, and strict ablations to substantiate the two-stage design.

**9JxX:** We justify slice-wise modeling and quantify inter-slice consistency through Z-Consistency, observing stable behavior without discontinuities.

**All changes are reflected in the revised manuscript with explicit references to new sections, tables, and figures.**

**Supporting Material:**

/attachment/208daa2e5a79c0154142ea26e70f90afb864d67c.pdf

---

### Meta-Review · Area_Chair_9TJ3 · 2026-02-09

**Recommendation:** Accept (Poster)
**Confidence:** 5

**Metareview:**

I thank the authors for their active participation in the rebuttal phase and responding to all the reviewer comments. Despite the mixed reviews I have decided to accept the submission, since the authors have managed to make significant improvements to the manuscript. Additionally, although I strongly agree with the concerns raised by the reviewers regarding using a private dataset, the fact that there is no suitable public dataset is the shortcoming of the field, and not of the submission. However, most departments would have easy access to such internal databases that could be used for implementing FluenceFormer. To make up for the lack of public datasets I would strongly urge the authors to make their code publicly available in the final submission. It would be of great benefit of the community, and concerns regarding generalization could be tested by others.

---

### Decision · Program_Chairs · 2026-02-14

Accept (Poster)